# Effects of Spatial Resolution on the Satellite Observation of Floating Macroalgae Blooms

Xinhua Wang [1,2,3], Qianguo Xing [1,2,3,*], Deyu An [4], Ling Meng [1,2,3], Xiangyang Zheng [1,2,3], Bo Jiang [1,2,3] and Hailong Liu [1,2,3]

1   CAS Key Laboratory of Coastal Environmental Processes and Ecological Remediation, Yantai Institute of Coastal Zone Research, Chinese Academy of Sciences, Yantai 264003, China; xinhuawang@yic.ac.cn (X.W.); lmeng@yic.ac.cn (L.M.); xyzheng@yic.ac.cn (X.Z.); bjiang@yic.ac.cn (B.J.); hlliu@yic.ac.cn (H.L.)
2   Center for Ocean Mega-Science, Chinese Academy of Sciences, Qingdao 266071, China
3   University of Chinese Academy of Sciences, Beijing 100049, China
4   Institute of Oceanographic Instrumentation, Qilu University of Technology (Shandong Academy of Sciences), Qingdao 266100, China; adylzl@163.com
*   Correspondence: qgxing@yic.ac.cn; Tel.: +86-535-210-9125

**Abstract:** Satellite images with different spatial resolutions are widely used in the observations of floating macroalgae booms in sea surface. In this study, semi-synchronous satellite images with different resolutions (10 m, 16 m, 30 m, 50 m, 100 m, 250 m and 500 m) acquired over the Yellow Sea, are used to quantitatively assess the effects of spatial resolution on the observation of floating macroalgae blooms of *Ulva prolifera*. Results indicate that the covering area of macroalgae-mixing pixels (MM-CA) detected from high resolution images is smaller than that from low resolution images; however, the area affected by macroalgae blooms (AA) is larger in high resolution images than in low resolution ones. The omission rates in the MM-CA and the AA increase with the decrease of spatial resolution. These results indicate that satellite remote sensing on the basis of low resolution images (especially, 100 m, 250 m, 500 m), would overestimate the covering area of macroalgae while omit the small patches in the affected zones. To reduce the impacts of overestimation and omission, high resolution satellite images are used to show the seasonal changes of macroalgae blooms in 2018 and 2019 in the Yellow Sea.

**Keywords:** floating macroalgae of *Ulva prolifera*; seasonal changes; macroalgae coverage; omission rate; satellite images; spatial resolution; the Yellow Sea

## 1. Introduction

Macroalgal blooms (MABs) caused by fast growth and accumulation of floating macroalgae have been increasing in global oceans in recent years [1–4]. These blooms can bring economic loss to human society as well as the changes in the marine ecosystem [5–8]. Quantitative monitoring of the floating macroalgae is essential for effective responses to floating MABs [9–12]. Various satellite optical sensors with different spatial resolutions have been used to monitor the widespread large-scale disaster, such as the widely used MODIS (Moderate Resolution Imaging Spectroradiometer), GOCI (Geostationary Ocean Color Imager), Landsat, HJ-1A/B, GF (GaoFen) and so on [13,14].

The macroalgae pixels detected from satellite images are mostly mixed by macroalgae and sea surface, which may bring uncertainties in the estimation of the coverage or the biomass [15–18]. The MODIS with resolutions of 250 m, 500 m and 1000 m can be used to detect large sizes of macroalgae, while the Sentinel-2 Multispectral Imager (MSI) and GF Wide Field of View (WFV) images, with the resolutions of 10 m and 16 m respectively, can be used to detect small ones [10,13,19–21]. The Moderate-resolution Wide-wavelengths Imager (MWI) is the ocean color sensor onboard the Chinese Tiangong-2 Space Lab, which was launched on 15 September 2016. The MWI is also an experimental satellite sensor

for the Chinese next generation ocean color satellites [22]. The MWI have 14 visible bands with a spatial resolution of 100 m, which may be also used to monitor floating macroalgae [23]. And, the CZI (Coastal Zone Imager) carried by the HY-1C satellite launched on 7 September 2018 is mainly used to monitor coastal zones [24]; the advantages of high-resolution (50 m) and short revisit period (3 days) of CZI make it convenient for observation of floating macroalgae.

Based on the similar reflectance features of vegetation, various indices are proposed for monitoring the floating macroalgae or algal blooms, NDVI, FAI [25], VB-FAH [13], DVI and other indices [26–29]. The band difference indices like FAI, VB-FAH, DVI shows linear responses to the portion of macroalgae on sea surface [30,31], and thus they are relatively robust for quantitative observation of floating macroalgae. The coverages of floating macroalgae monitored by multi-source images are significantly different, which is mainly caused by the mixed pixels of images with different spatial resolutions [32]. Currently, three indicators are commonly used to evaluate the monitoring results of floating macroalgae by satellite images: The covering area of macroalgae-mixing pixels (MM-CA) [9,10,33], the covering area of pure macroalgae (PM-CA) [11,34,35] and the affected area by macroalgae blooms (AA) [9]. Although the effects of resolution on the extraction of macroalgae pixels are noticed or studied with limited image pairs in some researches as the above-mentioned ones, how do these three indicators change with the resolution of remote sensing images, and how do they temporally and spatially change? The answers are very important for evaluating the results of MABs derived from satellite images with various resolutions.

In this study, the coverages of MM-CA, PM-CA and AA of macroalgae obtained by multi-source remote sensing images are compared, and the effects of satellite image spatial resolution on monitoring of floating macroalgae and the corresponding spatial and temporal characteristics are intensively investigated for promoting the quantitative assessment of floating macroalgae by satellite remote sensing.

## 2. Data and Methods

### 2.1. The Study Area

Figure 1 shows the study area. The dotted red line (Figure 1a) shows the major region affected by floating green macroalgae in summer [12]. The black box is the area monitored by the HY-1C CZI satellite images on 23 June 2019 (Figure 1b); the red slicks of macroalgae are extracted from the CZI images using dynamic threshold of DVI (see the Section 2.2). The in-situ photos and satellite images show that the size of floating macroalgae patches may vary from several centimeters to more than ten kilometers.

### 2.2. Satellite Images and Data Processing

Three groups of satellite images with various spatial resolutions with the presence of massive MABs over the Yellow Sea acquired on the same day are selected. The imaging time and spatial resolution of the images are shown in Table 1.

**Table 1.** Three groups (A, B and C) of semi-synchronous satellite images.

| Satellite Images | Sensing Date and Time | Spatial Resolution (m) |
|---|---|---|
| A1, Sentinel-2 MSI | 3 June 2018 10:36:51 | 10 |
| A2, Tiangong-2 MWI | 3 June 2018 14:30:33 | 100 |
| A3, Terra MODIS | 3 June 2018 11:10:00 | 250 |
| A4, Terra MODIS | 3 June 2018 11:10:00 | 500 |
| B1, GF-1 WFV | 23 June 2019 11:05:11 | 16 |
| B2, HY-1C CZI | 23 June 2019 11:43:25 | 50 |
| B3, Aqua MODIS | 23 June 2019 13:30:00 | 250 |
| B4, Aqua MODIS | 23 June 2019 13:30:00 | 500 |
| C1, Landsat-5 TM | 24 June 2009 10:06:30 | 30 |
| C2, Aqua MODIS | 24 June 2009 13:05:00 | 250 |
| C3, Aqua MODIS | 24 June 2009 13:05:00 | 500 |

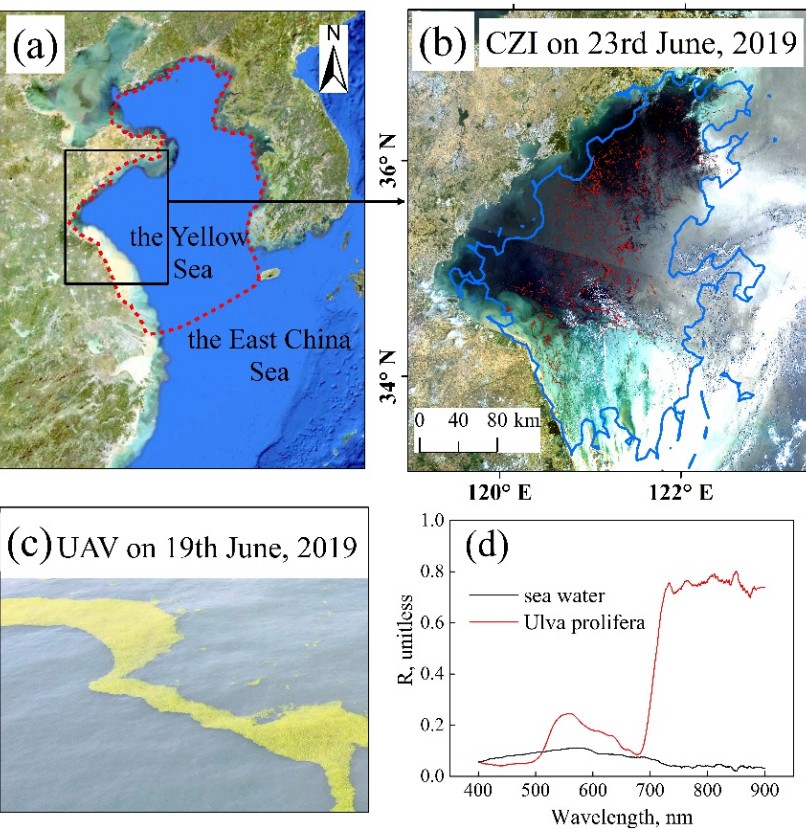

**Figure 1.** (**a**) Location of the study area; (**b**) The HY-1C CZI true color composite of band3 (red), band2 (green) and band1 (blue) acquired on 23 June 2019; (**c**) UAV photo of green tide of *Ulva prolifera* on 19 June 2019 (by Qianguo Xing); (**d**) The reflectance of sea water and *Ulva prolifera* measured on 19 June 2019 (by Bo Jiang and Hailong Liu).

HY-1C CZI images are calibrated by geometric correction and radiometric calibration, and the DN values are converted to reflectance (R, unitless) by Rayleigh correction. The FLAASH (Fast Line-of-sight Atmospheric Analysis of Spectral Hypercubes) atmospheric correction module is applied to GF-WFV, Landsat TM and MODIS images to obtain the reflectance (R, unitless), while the Sen2Cor and QUAC (Quickly Atmospheric Correction) atmospheric correction module are used to the Sentinel-2 MSI and Tiangong-2 MWI images, respectively.

The difference of vegetation index (DVI), which is linearly correlated with the coverage of floating macroalgae [11,13,36,37], is used to detect floating macroalgae in this work. The DVI images are derived as Equation (1).

$$DVI = R_{NIR} - R_{Red} \tag{1}$$

where $R_{NIR}$ is the reflectance at the near-infrared (NIR) band and $R_{Red}$ is the reflectance of Red band. The DVI images are segmented into small windows, and a set of thresholds which depends on the water surface optical conditions [11], are used to classify the macroalgae pixels window by window. And standard false color image is also used to visually inspect macroalgae patches i.e., if the DVI values higher than the threshold, then the pixels will be classified as macroalgae pixels [11]. Although different ways of atmospheric correction are applied to the images in this work, the results of extracted macroalgae coverages are generally consistent and comparable due to the dynamic thresholds in DVI used for the images with different atmospheric corrections. In this work, the "Explode Multipart Feature" tool in the Advanced Editing options of ArcMap is used to determine the macroalgae patch which is composed by one macroalgae pixel or a group of connected macroalgae pixels.

### 2.3. Estimation of the Area of Floating Macroalgae

A sub-pixel coverage model is used to estimate the covering area of pure macroalgae [37]. The portion of macroalgae (POM) in each pixel is indexed by its normalized DVI value [01DVI] as shown Equations (2) and (3). The covering area of pure macroalgae (PM-CA) is further calculated as Equation (4). The covering area of macroalgae-mixing pixels (MM-CA) is calculated as Equation (5).

$$POM = [01DVI] \tag{2}$$

$$[01DVI] = (DVI - DVI_{min})/(DVI_{max} - DVI_{min}) \tag{3}$$

$$PM\text{-}CA = \sum_{i=1}^{n} POM_i \times P_i \tag{4}$$

$$MM\text{-}CA = \sum_{i=1}^{n} P_i \tag{5}$$

where $DVI_{min}$ and $DVI_{max}$ are the minimum and the maximum values in a DVI image, and they are the DVI values of the macroalgae-free sea surface and pure macroalgae pixels, respectively; $P_i$ is the area ($km^2$) of the i th pixel containing macroalgae.

The area of the polygons enclosing all the macroalgae patches (Figure 1b), is set as the affected area by macroalgae blooms (AA). Then, the aggregation density (AD) of macroalgae (the ratio of MM-CA to AA), is calculated as Equation (6).

$$AD = MM\text{-}CA/AA \tag{6}$$

In this study, in each group of semi-synchronous satellite images, the zone with polygons enclosing all macroalgae pixels from the finest resolution image is set as the reference zone (i.e., $AA_0$) to assess the omission rate (OR, %) for other images with low resolutions, as Equation (7).

$$OR_{AA}, \% = (AA_0 - AA_i)/AA_0 \tag{7}$$

where $AA_0$ is the reference AA derived from Sentinel-2 MSI, GF-WFV or Landsat-5 TM image; and, $AA_i$ is the AA derived from the left images in each group. Similarly, as shown by Equation (8), the MM-CA is also used to evaluate the potential omission phenomenon in low resolution images.

$$OR_{MM\text{-}CA}, \% = (MM\text{-}CA_0 - MM\text{-}CA_{0i})/MM\text{-}CA_0 \tag{8}$$

where $MM\text{-}CA_0$ is the reference MM-CA derived from Sentinel-2 MSI, GF-WFV or Landsat-5 TM image; and, $MM\text{-}CA_{0i}$ is the $MM\text{-}CA_0$ of macroalgae pixels overlapped by the corresponding ones extracted from the left images in each group. To reduce the impacts caused by the changes in the positions of macroalgae patches, the spatial scopes of the macroalgae pixels of left images in each group are enlarged by a buffer zone of 0.5 km, and the $MM\text{-}CA_0$ of macroalgae pixels in the zones enclosed by the buffer zones ($MM\text{-}CA_{0i}$) is calculated.

## 3. Results and Analysis

### 3.1. Variations of MM-CA, AA and AD Derived from Different Resolution Images

For the semi-synchronous satellite images with different resolutions, the frequency and MM-CA of macroalgae patches in the same region monitored by satellites images are calculated, and the results are shown in Figure 2. The results show that, both the number of macroalgae patches and the total area of macroalgae pixels (MM-CA) tend to decrease with the increase in the patch size; and the total number of macroalgae patches decreases with the increase of image spatial resolution. In this case, the small size patches detected in Sentinel-2 MSI images (0~0.002 $km^2$) account for 94.8% of the total MM-CA, while 94.2% of macroalgae patches detected by Tiangong-2 MWI images are small patches (0~0.4 $km^2$); However, the large patches (>1 $km^2$) detected by the corresponding MODIS images with

spatial resolution of 250 m and 500 m have relatively higher portions in the total MM-CA, i.e., 41.2% and 73.5%, separately.

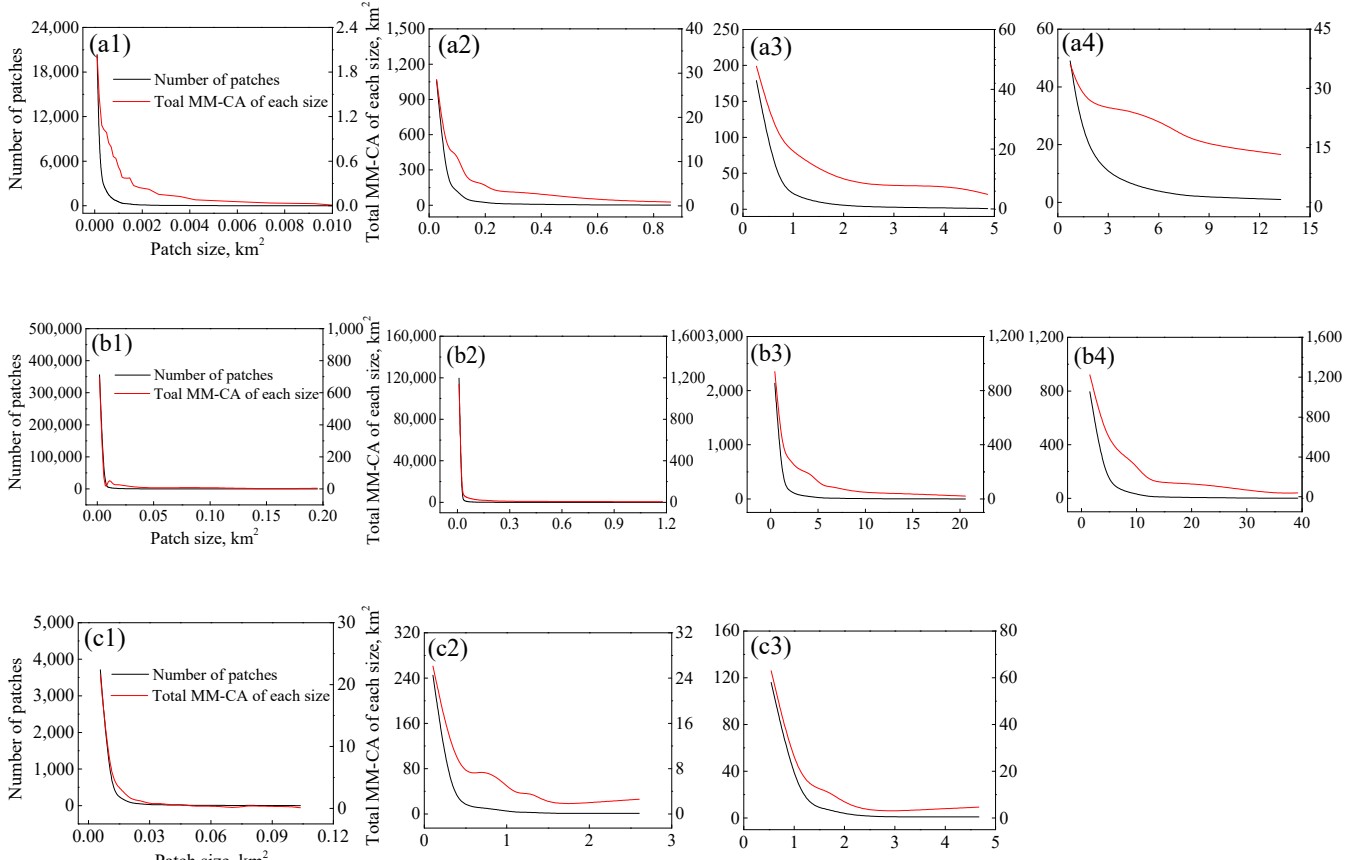

**Figure 2.** The number of patches and total MM-CA of different sizes. (**a1**–**a4**,**b1**–**b4**,**c1**–**c3**) are the images corresponding to A1–A4, B1–B4 and C1–C3 in Table 1, respectively.

Figure 3 shows the distributions of macroalgae patches and the affected zones derived from the three different groups of images (Table 1). The corresponding parameters of MM-CA, AA, AD and OR are calculated and shown in Table 2. For the images with the spatial resolution ranges from 10 m to 500 m, the MM-CA and the AD derived from high resolution images are smaller than those from low resolution images, while the affected areas (AA) derived from high resolution images are larger than those from low resolution images. The omission rates (OR) indexed by both the affected zones (AA) and the detected macroalgae pixels (MM-CA) increase with the decrease of image spatial resolution (Figure 4). Although the resolutions of reference images (Sentinel-2 MSI, GF1-WFV, Landsat-5 TM) are different, images with the resolutions larger than 100 m tend to omit considerable part of floating macroalgae by more than 50% or so in this study. In this study with floating MABs of *Ulva prolifera* in the Yellow Sea, when the magnitude indexed by MM-CA or AA is large, there is relatively large values of AD while small values of OR, for example, the group B of images acquired on 23 June 2019.

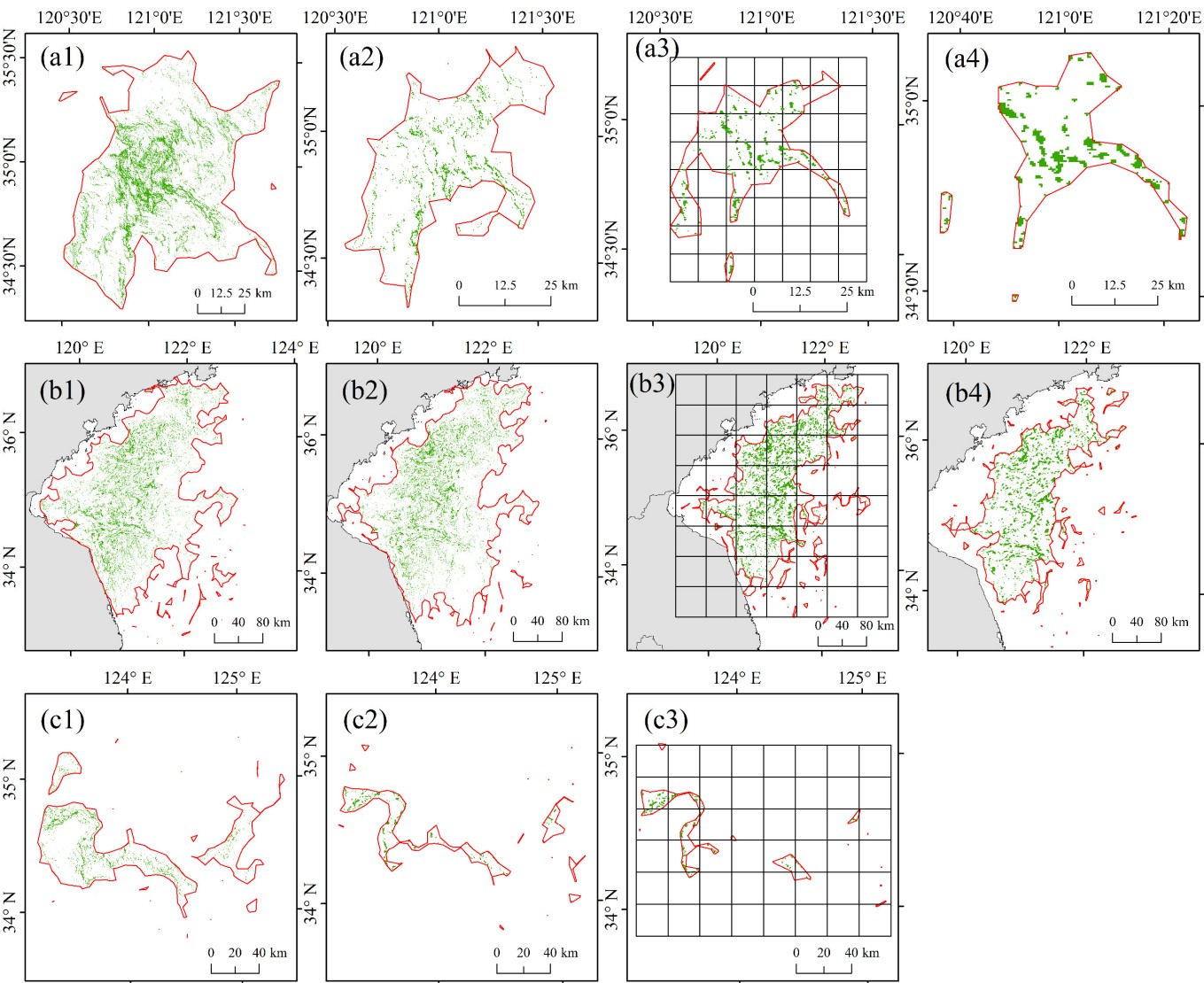

**Figure 3.** Comparison of spatial distribution of macroalgae obtained from images with different spatial resolutions. (**a1–a4,b1–b4,c1–c3**) are the images corresponding to those of A1–A4, B1–B4 and C1–C3 in Table 1, respectively.

**Table 2.** MM-CA, AA, AD, $OR_{AA}$ and $OR_{MM-CA}$ of macroalgae in three groups of images.

| Satellite Images | MM-CA (km$^2$) | AA (km$^2$) | AD | $OR_{AA}$ | $OR_{MM-CA}$ |
|---|---|---|---|---|---|
| A1. Sentinel-2 MSI (10 m) | 27.13 | 8243.24 | 0.33% | - | - |
| A2. TG-2 MWI (100 m) | 93.17 | 4073.31 | 2.29% | 51.98% | 47.71% |
| A3. Terra MODIS (250 m) | 108.83 | 2038.39 | 5.34% | 75.33% | 58.04% |
| A4. Terra MODIS (500 m) | 158.07 | 1089.39 | 14.51% | 86.78% | 64.09% |
| B1. GF-1 WFV (16 m) | 1584.75 | 59,728.31 | 2.65% | - | - |
| B2. HY-1C CZI (50 m) | 2293.66 | 55,235.83 | 4.15% | 11.21% | 9.53% |
| B3. Aqua MODIS (250 m) | 3103.49 | 35,327.71 | 8.78% | 41.42% | 48.82% |
| B4. Aqua MODIS (500 m) | 3644.28 | 27,658.22 | 13.18% | 53.84% | 49.33% |
| C1. Landsat-5 TM (30 m) | 29.26 | 6389.28 | 0.46% | - | - |
| C2. Aqua MODIS (250 m) | 78.34 | 1785.27 | 4.39% | 72.77% | 44.05% |
| C3. Aqua MODIS (500 m) | 96.01 | 956.84 | 10.03% | 85.3% | 50.34% |

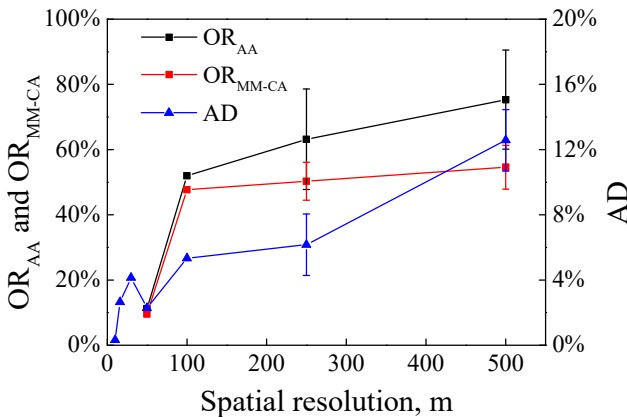

**Figure 4.** $OR_{AA}$, $OR_{MM-CA}$ and AD of macroalgae vary with spatial resolution.

As shown by Figure 3a3,b3,c3, the macroalgae-covered zones are empirically divided into some subzones and the MM-CA, the AA, the AD and the OR are calculated for each subzone. Significant decreasing trends in OR with the increase of AD are observed for all the three group images (Figure 5). The above results suggest that the omission of small patches in the satellite detections are more likely to occur when the aggregation density (AD) is low.

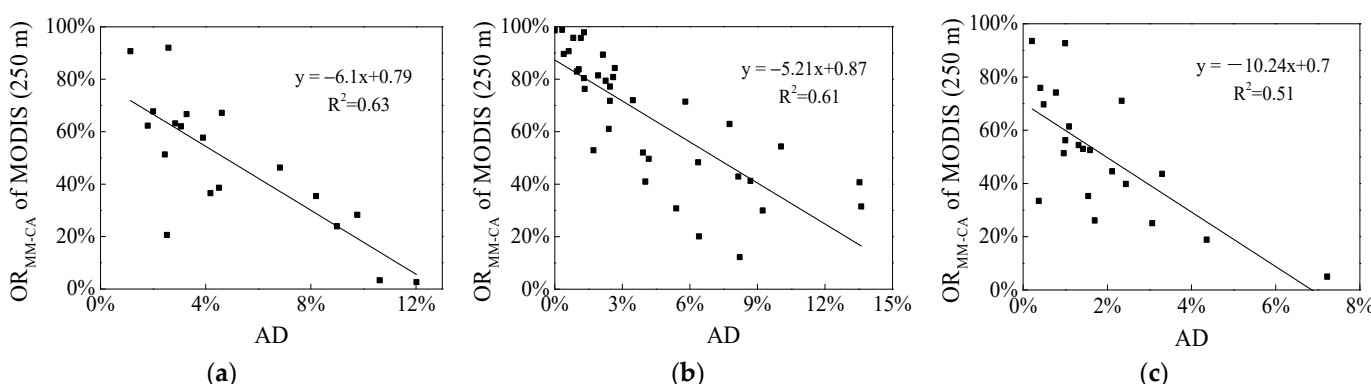

(**a**)　　　　　　　　　　　(**b**)　　　　　　　　　　　(**c**)

**Figure 5.** Changes in the omission rate ($OR_{MM-CA}$) of MODIS (250 m) image with the aggregation density (AD) of macroalgae in the subzones. (**a**–**c**) are corresponding to the group A–C, respectively.

On the basis of the MM-CA calculated for the above-mentioned subzones (as shown in Figure 3), the relationship in the MM-CA between the low resolution image and the corresponding high resolution image is analyzed. As shown by Figure 6, in general, there are good linear relationships between the MM-CA derived from different resolution images; and, compared with the high resolution images, the lower resolution tends to result in larger values in the MM-CA, i.e., more overestimation, which is due to the effects of mixed pixels of sea water and floating macroalgae. The slope of the fitting line may vary with the change in the spatial resolution of reference image; however, all the slopes increase with the spatial resolutions of themselves. For example, for the images with the resolutions of 100 m, 250 m and 500 m in group A, the slopes of the fitting lines are 2.99, 5.03 and 7.52, respectively.

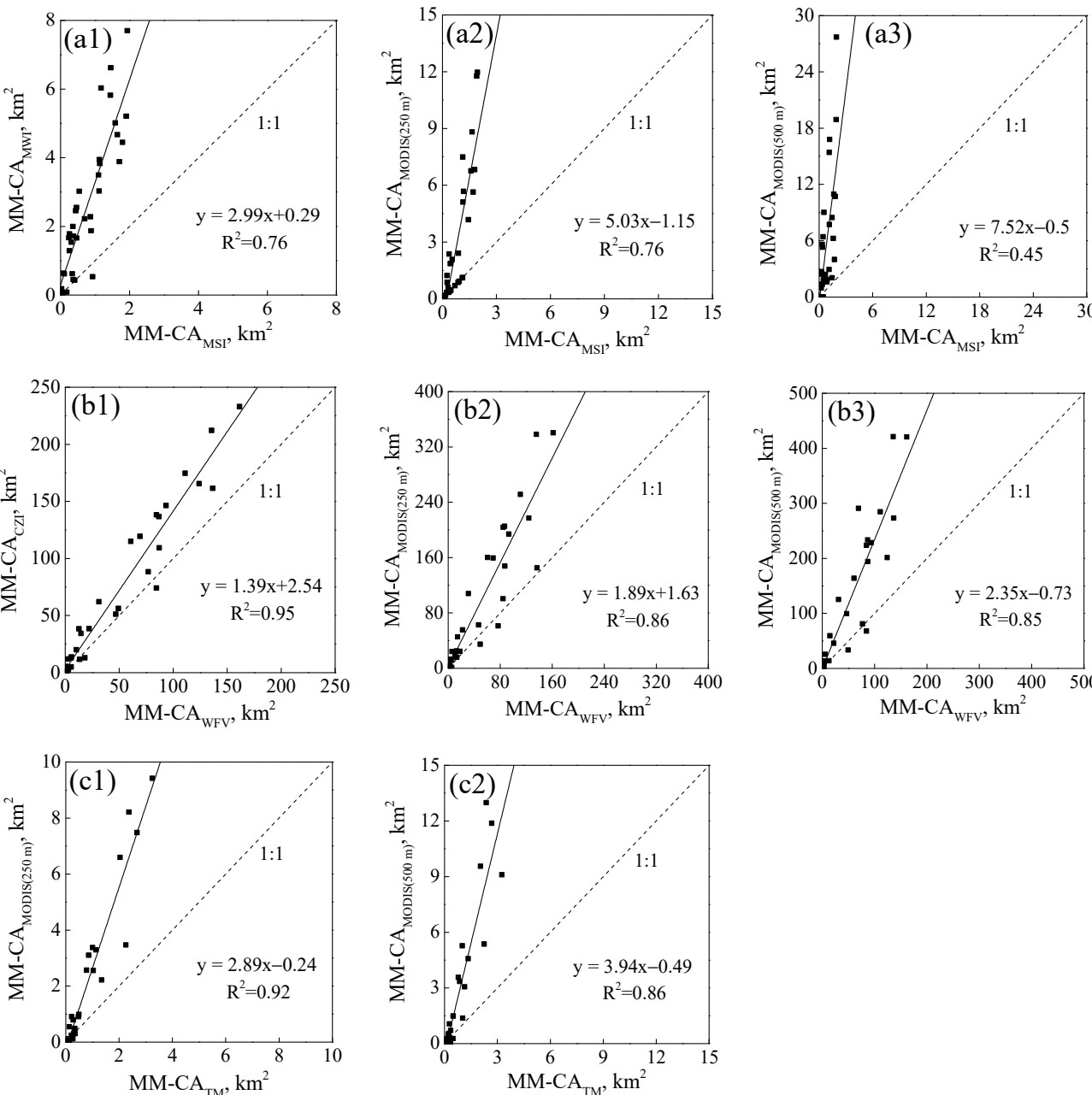

**Figure 6.** Relationship in the MM-CA derived from different resolution images in the subzones: (**a1**–**a3**,**b1**–**b3**,**c1**,**c2**), are corresponding to those of A2–A4, B2–B4 and C2,C3 in Table 1, respectively.

### 3.2. Seasonal Changes in Floating Macroalgae Coverages Estimated by High Resolution Images

Even in the high resolution images, such in Sentinel-2 MSI (10 m), GF-1 WFV (16 m) and Landsat-5 TM (30 m) in this study, most of the macroalgae pixels are mixed ones with various portions of macroalgae and sea surface. For high resolution images, in addition to the discussed macroalgae coverage parameters of MM-CA and AA, it is possible to estimate the area portion of pure macroalgae (POM) in each pixel and further the PM-CA. In this study, compared to Sentinel-2 MSI and Landsat-5 TM, multi-temporal images of GF-1 WFV (16 m) which has more data available for monitoring the entire growth cycle of macroalgae acquired over the Yellow Sea in 2018 and 2019 are used to estimate the macroalgae coverages.

The seasonal changes in MM-CA, PM-CA: MM-CA and AD of WFV images in 2018 and 2019 are shown in Figure 7. As shown by the MM-CA, the green tide in 2018 and 2019

shows similar patterns during the period from May to July, i.e., the MM-CA increased, reached its maximum in late June, and then it decreased until the disappearance of green tide in late July and early August. The temporal change patterns in the PM-CA: MM-CA and AD are similar to that of MM-CA, which means that the averaging densities of individual macroalgae patches also reach the maximums with the occurrence of the maximum macroalgae bloom.

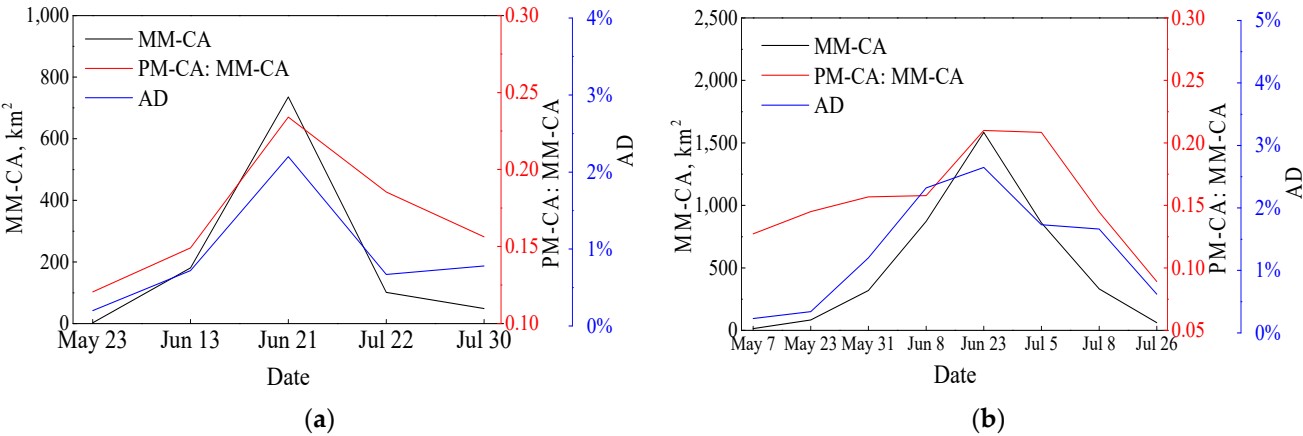

(**a**)　　　　　　　　　　　　(**b**)

**Figure 7.** Changes of MM-CA, PM-CA: MM-CA and AD of macroalgae derived from GF-1 WFV images in 2018 (**a**) and 2019 (**b**), respectively.

On the basis of the size, the macroalgae patches are divided into three groups: large patches (with the area of individual patch larger than $10^{-2}$ km$^2$), medium patches ($10^{-3} \sim 10^{-2}$ km$^2$) and small patches ($<10^{-3}$ km$^2$), and, the size compositions of macroalgae patches derived from GF-1 WFV images in 2018 and 2019 are shown in Figure 8. The results show that the small size patches ($<10^{-3}$ km$^2$) tend to decrease in number, which may be due to the growth of macroalgae during the expansion of green tide. The portion of large size patches usually reaches the highest value with the MM-CA.

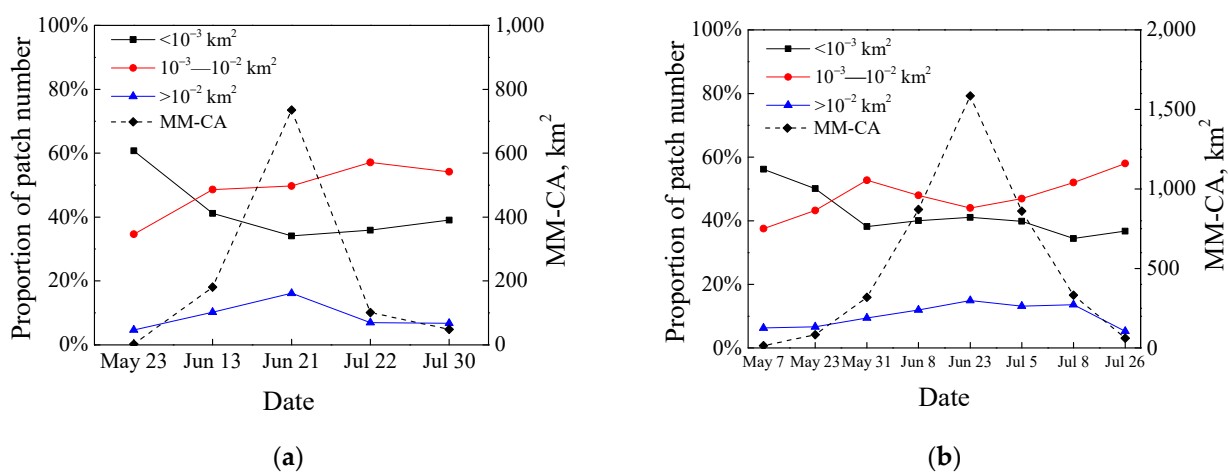

(**a**)　　　　　　　　　　　　(**b**)

**Figure 8.** Changes of proportion of patch number and MM-CA of macroalgae in different periods. (**a**) GF WFV images in 2018. (**b**) GF WFV images in 2019.

## 4. Conclusions and Prospect

Satellite images with different spatial resolutions can bring different results of covering areas in the observations of floating macroalgae booms. In this study, semi-synchronous satellite images with different resolutions, Sentinel-2 MSI (10 m), Gaofen WFV (16 m), Landat-5 TM (30 m), HY-1C CZI (50 m), Tiangong-2 MWI (100 m), and MODIS (250 m and 500 m) are used to extract the floating macroalgae blooms (MABs) of *Ulva prolifera* in the

Yellow Sea. Results suggest that satellite remote sensing on the basis of low resolution images (especially, 100 m, 250 m, 500 m), would lead to biased values by overestimating the covering area of macroalgae and omitting the small patches. Thus, high resolution satellite images of Gaofen-1 WFV (16 m), are used to extract the *Ulva prolifera* macroalgae blooms (MABs) in 2018 and 2019 in the Yellow Sea, and the seasonal patterns are shown.

This work suggest that the image resolution should be fully considered in the estimation of floating macroalgae, especially for spatial and/or temporal comparisons. It should be noted that this work is conducted over the Yellow Sea basin and with the floating macroalgae of *Ulva prolifera*; for other macroalgae, e.g., Sargassum [3,14], the effects of resolution may be different.

**Author Contributions:** Conceptualization, Q.X.; Image data processing and analysis, X.W. and D.A.; formal analysis, Q.X.; Field observations and spectral data processing, Q.X., H.L. and B.J.; writing—original draft preparation, Q.X., D.A. and X.W. ; writing—review and editing, Q.X., L.M. and X.Z.; funding acquisition, Q.X. All authors have read and agreed to the published version of the manuscript.

**Funding:** This study was partly supported by the National Natural Science Foundation of China (Nos 42076188, 41676171 and 41911530237) the Chinese Academy of Science Strategic Priority Research Program—the Big Earth Data Science Engineering Project (Nos XDA1906000, XDA19060203 and XDA19060501); the Instrument Developing Project of the Chinese Academy of Sciences (No. YJKYYQ20170048).

**Institutional Review Board Statement:** Not applicable.

**Informed Consent Statement:** Not applicable.

**Data Availability Statement:** According to the requirement of confidentiality agreement, the data used in this paper is not public.

**Acknowledgments:** The authors are thankful to the anonymous reviewers for their useful suggestions.

**Conflicts of Interest:** The authors declare no conflict of interest.

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
