# Peer review of "Effects of Spatial Resolution on the Satellite Observation of Floating Macroalgae Blooms"

_water, doi:10.3390/w13131761_

Round 1

Reviewer 1 Report

The manuscript „Effects of Spatial resolution on the Satellite Observation of Floating Macroalgae Blooms“ compares satellite images of Ulva prolifera bloom with different resolution. The manuscript is clearly written and well readable and understandable. I have only a few notes:

1/  Line 180: “resolutions of 100 m, 250 m and 250 m” – is it OK? It should be “100 m, 250 m, and 500 m”, should not be?

2/ Ulva prolifera – the Latin name of the algae should be in Italica

3/ Line 31: the abbreviation MABs is introduced but it is not used through the text. It is often replaced by the full text “macroalgae blooms” (e.g. in Conclusions at least three times)

Reviewer 2 Report

This study used multi-source satellite data with different spatial resolutions to detect floating macroalgae in the Yellow Sea and examined the impact of spatial resolutions on the detection results. My primary concern is that similar research and findings have been reported in the literature, e.g., Xu, Qing, Hongyuan Zhang, and Yongcun Cheng, Multi-sensor monitoring of Ulva prolifera blooms in the Yellow Sea using different methods, Frontiers of Earth Science 10, no. 2 (2016): 378-388. This is simply an example and the authors may directly state in the introduction how this study is innovative and advances previous research.

My specific comments are shown below:

  1. Lines 74-75, “Figure 1b, the patches in red are the floating macroalgae”. How the red patched were identified? Or, how AA on Line 109 was determined?
  2. Line 81, “measured by on 19th June, 2019”, please correct the typo.
  3. Why using DVI to detect floating macroalgae? How about other indices? Do they generate similar results in this work?
  4. The authors used Rayleigh corrected reflectance of HY-1C CZI data. For other satellite data, FLAASH, Sen2COR, and QUAC were used. First, FLAASH and Sen2COR were not designed for water, there may be substantial errors in retrieving reflectance of water surface. Second, for each satellite image after atmospheric correction, there is no evaluation in terms of the atmospheric correction accuracy/data quality. Third, for satellite imagery processed by different atmospheric correction methods, are their reflectance comparable with each other? This is important because in this work, DVI is used (unlike NDVI which is a relative measure).
  5. Lines 130-138, I am curious how a “patch” was determined?
  6. What is the size of each subzone? Does the size of subzone affect the relationships in Figure 5? May perform some sensitivity analysis.
  7. Please make all the plots in Figure 6 square, not rectangle, as the x and y axis have the same unit and range.
  8. Section 3.2, why not use Sentinel-2 MSI with the best spatial resolution (10 m)?

Round 2

Reviewer 2 Report

The authors addressed all of my comments carefully. The quality of the manuscript was greatly improved.